# Depression in association with neutrophil-to-lymphocyte, platelet-to-lymphocyte, and advanced lung cancer inflammation index biomarkers predicting lung cancer survival

Barbara L. Andersen[1], John Myers[2], Tessa Blevins[1], Kylie R. Park[1], Rachel M. Smith[2], Sarah Reisinger[3], David P. Carbone[3], Carolyn J. Presley[3], Peter G. Shields[3], William E. Carson[4]*

1 Department of Psychology, Ohio State University Comprehensive Cancer Center, The Ohio State University, Columbus, Ohio, United States of America, 2 Department of Biomedical Informatics and Center for Biostatistics, College of Medicine, The Ohio State University, Columbus, Ohio, United States of America, 3 Department of Internal Medicine, Division of Medical Oncology, College of Medicine, Ohio State University Comprehensive Cancer Center, The Ohio State University, Columbus, Ohio, United States of America, 4 Department of Surgery, Division of Surgical Oncology, College of Medicine, Ohio State University Comprehensive Cancer Center, The Ohio State University, Columbus, Ohio, United States of America

☯ These authors contributed equally to this work.
‡ These authors also contributed equally to this work
* william.carson@osumc.edu

## Abstract

Lung cancer is a product of inflammation and a dysfunctional immune system, and depression has similar dysregulation. Depression disproportionately affects lung cancer patients, having the highest rates of all cancers. Systemic inflammation and depression are both predictive of non-small cell lung cancer (NSCLC) survival, but the existence and extent of any co-occurrence is unknown. Studied is the association between systemic inflammation ratio (SIR) biomarker levels and patients' depressive symptoms, with the hypothesis that depression severity would be significantly associated with prognostically poor inflammation. Newly diagnosed stage-IV non-small cell lung cancer (NSCLC; $N = 186$) patients were enrolled (ClinicalTrials.gov Identifier: NCT03199651) and blood draws and depression self-reports (Patient Health Questionnaire-9) were obtained. For SIRs, cell counts of neutrophils (N), lymphocytes (L), and platelets (P) were abstracted for ratio (R) calculations for NLR, PLR, and the Advanced Lung cancer Inflammation Index (ALI). Patients were followed and biomarkers were tested as predictors of 2-year overall survival (OS) to confirm their relevance. Next, multivariate linear regressions tested associations of depression with NLR, PLR, and ALI. Overall 2-year mortality was 61% (113/186). Cox model analyses confirmed higher NLR [hazard ratio (HR) = 1.91; $p = 0.001$] and PLR (HR = 2.08; $p<0.001$), along with lower ALI (HR = 0.53; $p = 0.005$), to be predictive of worse OS. Adjusting for covariates, depression was reliably associated with biomarker levels ($p \leq 0.02$). Patients with moderate/severe depressive symptoms were 2 to 3 times more likely to have prognostically poor biomarker levels. Novel data show patients' depressive symptoms were reliably associated with lung-relevant systemic inflammation biomarkers, all assessed at diagnosis/pretreatment. The

**Data Availability Statement:** All relevant data are within the paper and its Supporting Information files.

**Funding:** This study was funded by the Ohio State University Comprehensive Cancer Center and Pelotonia through grants awarded to PS and DC. The funders had no role in study design, data collection and analysis, decision to publish, or preparation of the manuscript.

**Competing interests:** BLA, JM, TB, KRP, RMS, SR, and WEC have no competing interests to declare. DPC reports personal fees from Abbvie, Adaptimmune, Agenus, Amgen, Ariad, AstraZeneca, Biocept, Boehringer Ingelheim, Celgene, Clovis, Daiichi Sankyo, Inc. (DSI), EMD Serono, Flame Biosciences, Foundation Medicine, G1Therapeutics/Intellisphere, GenePlus, Genentech/Roche, Glaxo-Smith-Kline, Gloria Biosciences, Gritstone, Guardant Health, Humana, Incyte, Inivata, Inovio, Janssen, Kyowa Kirin, Loxo Oncology, Merck, MSD, Nexus Oncology, Novartis, Oncocyte, Palobiofarma, Pfizer, prIME Oncology, Stemcentrx, Takeda Oncology, and Teva; and grants and personal fees from Bristol Myers-Squibb (BMS) outside the submitted work during the conduct of the study. This does not alter our adherence to PLOS ONE policies on sharing data and materials.

same SIRs were found prognostic for patients' 2-year OS. Intensive study of depression, combined with measures of cell biology and inflammation is needed to extend these findings to discover mechanisms of depression toxicity for NSCLC patients' treatment responses and survival.

## Introduction

Immunotherapy and targeted treatments are yielding significantly longer survival for patients with advanced (stage IV) non-small cell lung cancer (NSCLC) [1–3], but it is unknown why a substantial portion of patients fail to respond to treatment. Causes are multifactorial, but an unexplored contributor to premature death may be toxic, biobehavioral (psychological, behavioral, biological) characteristics of co-morbid depression.

Several lines of evidence point to the plausibility of depression influencing disease progression. In the general case, depression assessed at diagnosis is prognostic for mortality in cancer patients [4], with the strongest effects found in lung cancer [5]. Moreover, the *continuing trajectory* of depressive symptoms—from diagnosis through two years—predicts overall NSCLC survival [6]. Discovery of such effects is made possible, unfortunately, because of the high incidence of depression in NSCLC patients. Of all cancer patients, those with lung cancer are among the most impaired [5,7,8], with an estimated 36% of NSCLC with moderate to severe depressive symptoms at diagnosis, many of whom also have comorbid anxiety [9], a clinical presentation more resistant to psychological treatment [10].

In addition to psychologic aspects, depression's biologic dysregulation may contribute to disease progression. Lung cancer is a product of a dysfunctional immune system, as evidenced by tobacco/smoking-induced inflammation [11], an inflammatory tumor microenvironment [12], and robust, systemic inflammatory responses (SIRs) [12,13]. In this context, there is "overwhelming" evidence [14] that severe depressive symptoms and major depressive disorder (MDD) covary with elevations of proinflammatory cytokines [15], decreased adaptive immune responses [16–19], and others [20]. Further, biologic therapies which increase inflammation (e.g. interferon-alfa treatment) can cause MDD [21,22]. Thus, studies suggest that inflammation with depression co-occurs with that arising from lung cancer.

Study of the health consequences of depression in lung cancer patients is timely as new therapies come on line and treatment guidelines rapidly change [23]. Psychosocial studies of lung patients have been few and come largely from prior decades of chemotherapy-only treatments [24], making examination association of depressive symptoms and systemic inflammation responses (SIRs) novel. Also important is discovery of the nature and biologic mechanisms of depressive symptom toxicity; depression is a rate limiting factor impacting patients' quality of life and, potentially, treatment response.

To do this, biomarkers of inflammation were studied in relationship to depressive symptoms. Systemic inflammatory responses (SIRs)—neutrophils, lymphocytes, monocytes, and platelets—are key factors in proliferation, angiogenesis, immunosuppression, and nutritional depletion [25]. Briefly, the inflammatory response is characterized by increases in circulating neutrophils (N) accompanied by falls in circulating lymphocytes (L). The neutrophil-to-lymphocyte ratio (NLR) is viewed as reflecting the inflammatory imbalance of pro-tumor efficacy (N) and anti-tumor capacity (L) of the host [25,26]. The platelet-to-lymphocyte ratio (PLR) is important as platelet elevation accelerates tumor progression [27,28]. Unique is the advanced lung cancer inflammation (ALI) index, which considers nutritional status with NLR. SIRs are

prognostic biomarkers for multiple tumor types and multiple meta analyses have confirmed elevated NLR [29] and PLR [29] and lower ALI [29] levels at diagnosis to predict NSCLC overall survival (OS).

Newly diagnosed NSCLC patients (N = 186) enrolled in an observational cohort (NCT03199651) participated. Procedures included routine laboratory work and patient-reported depressive symptom assessment [Patient Health Questionnaire-9 (PHQ-9)] [30,31]. First, NLR, PLR, and ALI were validated as predictors of OS at 2-years. If confirmed, associations of PHQ-9 scores with SIRs were next examined, with the hypothesis that depression severity would be significantly associated with higher NLR and PLR and lower ALI levels. Individually, patients with moderate to severe depressive symptoms were expected to have prognostically poor levels of systemic inflammation in comparison to patients without depressive symptoms. If confirmed, the findings would provide support for biobehavioral aspects of depression potentially "adding" to the inflammatory dysregulation already present in those with advanced NSCLC.

## Materials and methods

### Study design

A prospective cohort design was used. An NCI-designated Comprehensive Cancer Center enrolled advanced (stage IV) NSCLC patients at diagnosis (NCT03199651) and followed patients up to 5 years, withdrawal, or death. Accrual occurred from July 2017 to February 2020.

### Procedures

A university institutional review board approved the study and procedures in accordance with ethical standards and the 1964 Helsinki declaration. Inclusion criteria were as follows: newly diagnosed with pathology confirmed stage IV NSCLC (any type); any Eastern Cooperative Oncology Group (ECOG) performance status; any illness/condition comorbidity; age $\geq$ 18 years; English-speaking; willingness to provide access to medical records, provide biospecimens, and respond to patient reported outcome (PRO) measures. Exclusion criteria were as follows: prior treatment with definitive chemo-radiotherapy for lung cancer; diagnosis >90 days prior to enrollment; receipt of lung cancer treatment for over one month; presence of disabling hearing, vision, or psychiatric conditions (e.g., psychosis) preventing consent or completion of PROs in English. Also excluded were second opinion or consult only cases (patients returning to his/her community physician for treatment).

Research personnel completed informed consent face-to-face (verbal, with written materials) during a patient's first clinic appointment with a thoracic oncologist and written informed consent was obtained and the signature witnessed. Cell count and albumin data for biomarker calculations were retrieved from electronic medical record reports. Lab data on or immediately after the patients' first visit (mean = 11.82 days) were used. Interviewers from a professional survey center contacted patients by telephone and assessed PROs, for which patients received $15. Study flow is provided (Fig 1).

### Measures

**Depressive symptoms.** The American Society of Clinical Oncology's recommended measure was used, the Patient Health Questionnaire-9 (PHQ-9) [30,31]. Nine items assessing symptoms of major depressive disorder (MDD) are rated on a Likert frequency scale, ranging from 0 = not at all to 3 = nearly every day. Items are summed for a total score ranging from 0

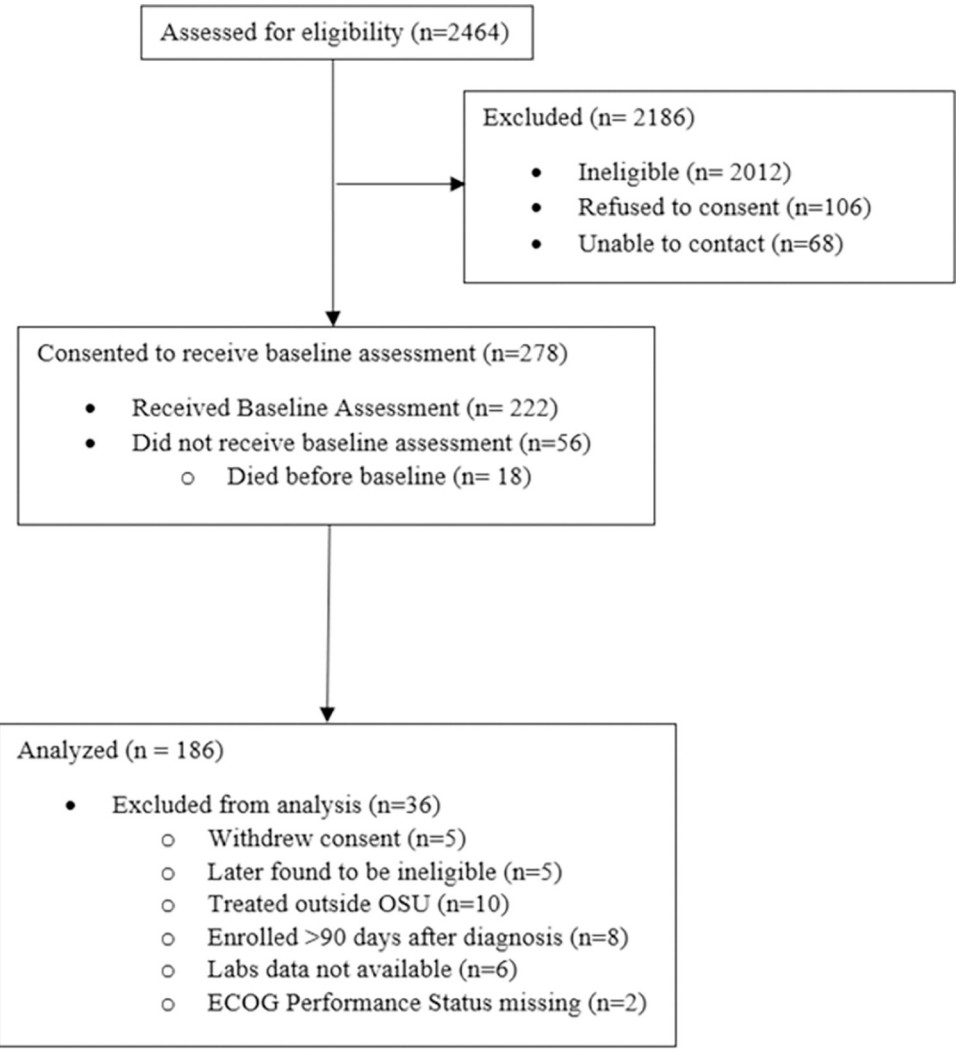

**Fig 1. Study flow.**

to 27. Symptom level classifications are as follows: 0–7 (none/mild), 8-14 (moderate), 15–19 (moderate to severe), and 20–27 (severe). Cronbach's alpha was .80.

**Systemic inflammation biomarkers.** 1) Neutrophil-to-lymphocyte ratio (NLR) was calculated by dividing a patient's absolute neutrophil count by the absolute lymphocyte count. 2) Platelet-to-lymphocyte ratio (PLR) was calculated by dividing platelet count ($10^9$/L) by the absolute lymphocyte count. 3) Advanced lung cancer inflammation index (ALI) was calculated by multiplying body mass index (BMI) by the quotient of albumin (g/dL) and NLR (i.e., ALI = BMI × Albumin/NLR), where BMI = weight (kg)/height (m)$^2$. Considering methods and findings from prior studies and meta analyses of SIRS as predictors of OS, the following accepted cutoffs were used for the survival analyses: 5 for NLR [32,33], 185 for PLR [34], and 24 for ALI [35].

**Covariates.** Variables considered were as follows: age, sex, race, partner status, education level, employment status, and smoking history. Also considered were lung cancer type, performance status, and first line treatment to be received as abstracted from medical records. The Eastern Cooperative Oncology Group Performance Status (ECOG PS) [36] score was assigned

by the treating physician. ECOG PS can range from 0 to 5: 0 = fully active; 1 = restricted in physically strenuous activity but ambulatory and able to carry out light work, 2 = ambulatory and capable of all self-care, but unable to work, 3 = capable of only limited self-care, confined to bed or chair more than 50%, 4 = completely disabled, confined to bed or chair, and, 5 = deceased.

## Power

*A priori* power analyses estimated the sample size necessary to detect a significant association of PHQ-9 scores, whether measured as continuous or categorical, with SIRs at baseline. Based on a significance level of alpha = .05, a sample size of 91 would yield an estimated power equal to 0.80 in linear regression with the six parameters and a medium effect size (Cohen's $F^2$ = 0.15). Sample size N = 186 translates to power >.95. This sample size is equivalent to or larger than those in meta analyses of the relationship between these biomarkers and NSCLC OS [29].

## Statistical analysis

Demographic and disease characteristics of the sample were summarized and scores on depression and biomarker ratio levels and cutoffs provided. In order to avoid overfitting the survival models, covariates for analysis were selected based on a cutoff of $p < 0.20$ from univariate models S1 and S2 Tables. Additionally, smoking status and treatment type were included in survival models to account for their effect on survival. Demographic variables such as age, sex, race, and marital status were selected as covariates for multiple linear regressions. Baseline ECOG Performance Status was also selected to serve as a proxy for patient fitness. A cut-off of $p < 0.20$ was used to select additional covariates for multiple linear regressions. Cox proportional models were used to assess the association between biomarkers as defined by cut-off levels and overall survival (OS), adjusting for relevant covariates.

Next, multiple linear regressions tested the association of patient depressive symptoms (PHQ-9) with elevated NLR, elevated PLR, and lower ALI, adjusting for relevant covariates. NLR, PLR, and ALI were log-transformed due to right-skewed distributions. Descriptive analyses show the percentage of patients with no/low versus high depressive symptoms stratified by low versus high SIR levels. A p-value of 0.05 was used as a cutoff for significance. Analyses were performed using R 4.1.1. [37].

## Results

### Sample description

Baseline demographics, smoking status, lung cancer type, and performance status for N = 186 are provided (Table 1). Characteristic of advanced NSCLC patients, the sample was older (mean 63 years), unemployed (74.7%), previously/currently smoking (84.9%), with the majority having either adenocarcinoma or squamous histologic subtype (89.8%). Patients were diagnosed a median of 34 days prior to the baseline assessment. The majority of patients were subsequently treated (91%), with chemotherapy alone (30%), immunotherapy alone (22%), chemotherapy+immunotherapy (19%), targeted therapy for mutations (e.g., EGFR, ALK) (16%), targeted therapy chemotherapy (2%), or other treatment (1%).

The median value for the PHQ-9 was 5.5 (Q1 = 3, Q3 = 9), corresponding to a mild level of depressive symptoms for the sample. Using classifications for individuals, 64.5% of patients reported depressive symptoms at the none/mild level (n = 120), 27.4% at the moderate level (n = 51), 4.8%% at the moderate to severe level (n = 9), and 3.2% at the severe level (n = 6).

**Table 1. Demographic and disease characteristics of NSCLC sample (N = 186).**

| Variable | NSCLC (N = 186) |
|---|---|
| | Mean (SD) or N (%) |
| **Demographics[a]** | |
| Age | 63.25 (10.66) |
| Race | |
| Caucasian | 154 (82.8%) |
| African American | 9 (4.8%) |
| Asian | 1 (0.5%) |
| Multiracial | 21 (11.3%) |
| Other | 1 (0.5%) |
| Ethnicity (% Hispanic) | 3 (1.6%) |
| Sex (% Male) | 110 (59.1%) |
| Marital Status (% married/partnered) | 110 (59.1%) |
| Education | |
| Less than High School | 22 (11.8%) |
| High School/GED | 64 (34.4%) |
| Greater than High School | 100 (53.8%) |
| Income (annual) (n = 170) | |
| ≤$15,000 | 18 (10.6%) |
| $15,001-$25,000 | 20 (11.8%) |
| $25,001-$50,000 | 60 (35.3%) |
| $50,001-$75,000 | 23 (13.5%) |
| $75,001-$100,000 | 20 (11.8%) |
| ≥$100,000 | 29 (17.1%) |
| Employment Status (% Not Employed) | 139 (74.7%) |
| BMI Category | |
| Underweight or Healthy weight | 75 (40.3%) |
| Overweight | 54 (29.0%) |
| Obese | 57 (30.6%) |
| **Lung risk factors and disease** | |
| Smoking (% Ever) | 158 (84.9%) |
| Cell Type | |
| Adenocarcinoma | 144 (77.4%) |
| Squamous | 23 (12.4%) |
| Adenosquamous | 7 (3.8%) |
| Large Cell | 4 (2.2%) |
| Not Otherwise Specified / Other | 8 (4.3%) |
| Treatment (% Subsequently receiving treatment) | 169 (90.9%) |
| ECOG-PS | |
| 0 | 68 (36.6%) |
| 1 | 94 (50.4%) |
| 2 | 22 (11.8%) |
| 3 | 2 (1.1%) |

Complete data is provided for all variables excepting annual household income (n = 170, 16 decline/missing observations).

SIR biomarker median and quartiles (Q1, Q3) were as follows: NLR 5.39 (Q1 = 3.33, Q3 = 10.70), PLR 224.45 (Q1 = 149.89, Q3 = 341.84), and ALI 18.24 (Q1 = 9.17, Q3 = 34.50). Using the cutoff values (NLR≥5, PLR≥185, and ALI≥24), the majority of the sample had elevated NLR (53%) and PLR (63%). Likewise, 62% of patients had high inflammation as indexed by low ALI.

## SIR biomarkers as prognostic for NSCLC overall survival (OS)

The median follow-up for the sample was 8 months, with an inter-quartile range of 3 to 22 months and a range of 0 to 24 months. Overall mortality at 2-years was 61% (113/186). Survival analyses used the SIR cutoffs noted above and adjusted for covariates (i.e., ECOG-PS, age, educational level, smoking status, and treatment received at baseline). Analyses determined statistically significant associations between each biomarker and overall survival (Table 2).

After adjusting for covariates, the hazard ratio (HR) for NLR is 1.91 (95% CI: 1.29, 2.84; p = 0.001). That is, patients with elevated NLR at diagnosis (higher inflammation) were approximately twice as likely to die at any time point compared to those with a lower inflammation ratio. Patients with baseline NLR<5 (47%) had an estimated one-year survival probability of 0.76 (95% CI: 0.67, 0.85), while those with NLR≥5 (53%) had a one-year survival probability of 0.46 (95% CI: 0.37, 0.57). The log-rank test was significant (p-value = 0.001) indicating a marked difference in survival (Fig 2).

The hazard ratio for elevated PLR is 2.08 (95% CI: 1.34, 3.22; p = 0.001), comparable to the association between NLR and survival. Patients with baseline PLR<185 (43%) had a one-year survival probability of 0.78 (95% CI: 0.69, 0.89), while those with a baseline PLR≥185 (57%) had a one-year survival probability of 0.49 (95% CI: 0.41, 0.59). The log-rank test was significant (p = 0.005), indicating a significant difference in survival between groups (Fig 3).

The hazard ratio for ALI is 0.53 (95% CI: 0.34, 0.82; p = 0.005), indicating that those with higher levels of ALI (i.e., indicative of lower inflammation) had better survival. Patients with baseline ALI<24 (62%) were approximately twice as likely to die than those with higher ALI, having an estimated survival probability of 0.50 (95% CI: 0.41, 0.60). Patients with baseline ALI≥24 (38%) had a one year estimated survival probability of 0.77 (95% CI: 0.77, 0.87). The log-rank test indicated a significant difference in survival (p < 0.005) between groups (Fig 4).

In summary, the relevance of the three SIRs was confirmed, with higher levels of inflammation predicting lower OS in this NSCLC sample.

**Table 2. Associations between systemic inflammation ratios (SIRs) and overall survival.**

| Inflammation Biomarker | Model | HR | 95% CI | *p*-value |
|---|---|---:|---|---:|
| NLR ≥ 5 | Unadjusted Model | 1.88 | 1.28, 2.75 | 0.001 |
| | Adjusted model | 1.91 | 1.29, 2.84 | 0.001 |
| PLR ≥ 185 | Unadjusted Model | 1.93 | 1.28, 2.92 | 0.002 |
| | Adjusted model | 2.08 | 1.34, 3.22 | 0.001 |
| ALI ≥ 24 | Unadjusted Model | 0.49 | 0.32, 0.74 | 0.001 |
| | Adjusted model | 0.53 | 0.34, 0.82 | 0.005 |

NLR, Neutrophil to Lymphocyte Ratio; PLR, Platelet to Lymphocyte Ratio; ALI, Advanced Lung Cancer Inflammation Index; HR, hazard ratio, CI, confidence interval.

NOTE: Associations are between inflammation biomarkers and overall survival. HR indicates the difference between biomarkers above and below the specified cutoff levels. The NLR and PLR models are adjusted for ECOG-PS (dichotomized as 0–1, 2 or more), age (dichotomized as <65, 65≥), educational level (dichotomized as no college education, at least some college), smoking status, and treatment received at baseline.

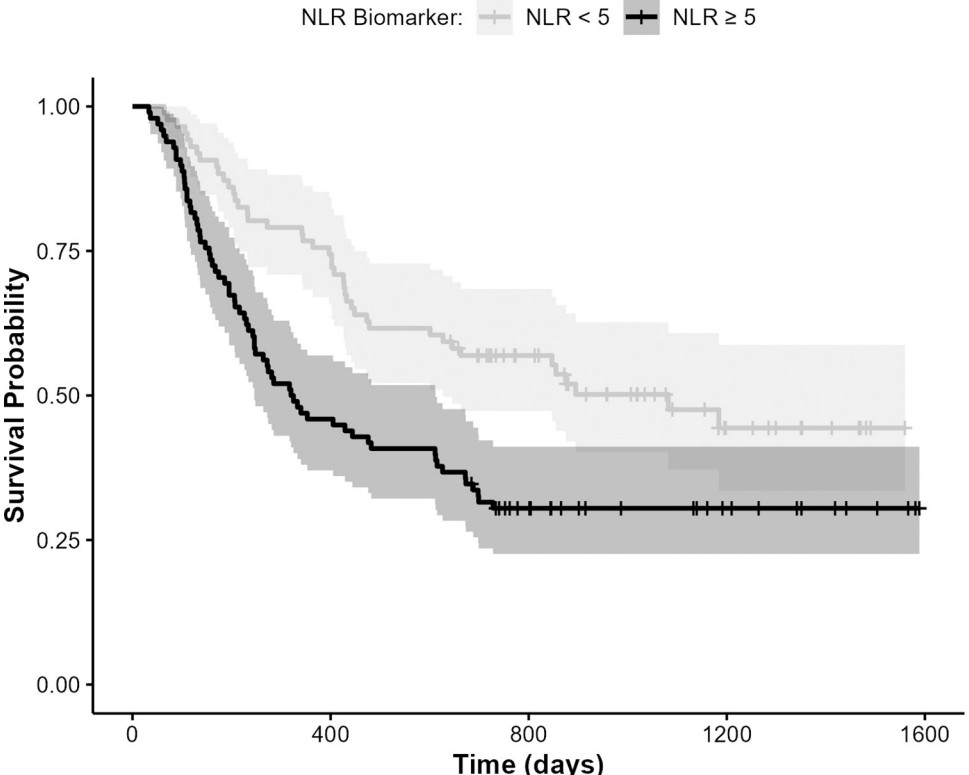

**Fig 2. Kaplan Meier survival curve for NSCLC patients stratified by NLR systemic inflammation, NLR < 5 and NLR ≥ 5 (n = 184), showing worse survival probability (p = 0.001) for patients having greater inflammation (i.e., above NLR cutoff).**

## Relationship of depressive symptoms with SIR biomarkers

Analyses supported the prediction of significant relationships between patients' depressive symptoms and SIRs, both obtained at diagnosis, adjusted for covariates (Table 3). That is, the PHQ-9 had a significant positive association with NLR (p = 0.02) and PLR (p = 0.02). In both cases, elevations in depressive symptoms were associated with greater elevations of neutrophils and platelets in relationship to lymphocytes. Also, PHQ-9 had an inverse, significant association with ALI (p = 0.009). High levels of depressive symptoms were associated with lower ALI values.

To illustrate significant associations at the patient level, figures display the relationship between depression [PHQ-9<8 (none/mild) vs ≥8 (moderate to severe)] and SIRs (e.g., NLR<5 vs. NLR ≥5), yielding 4 patient groups determined by cutoffs. For patients with no/ mild depressive symptoms, they were as likely to be found above (50%) as they were to be below (50%) the NLR cutoff (Fig 5, left side). Although not 50/50, similar percentages of patients with no/mild depressive symptoms were found above (56%) versus below (42%) the PLR cutoff (Fig 6, left side) and below (58%) versus above (42%) the ALI cutoff (Fig 7, left side).

In contrast, patients with moderate to severe depressive symptoms had, disproportionately, high inflammation rather than low, an effect seen across SIRs. For the depressed, 58% were above the NLR cutoff versus 42% below the cutoff (Fig 5, right side). The discrepancy is 2 to 3 times greater for PLR and ALI ratios. For the depressed, 77% were above the PLR cutoff versus 23% below (Fig 6, right side). The effect is the same, though reversed, for the depressed patients and ALI; 70% were below the cutoff verses 30% above (Fig 7, right side).

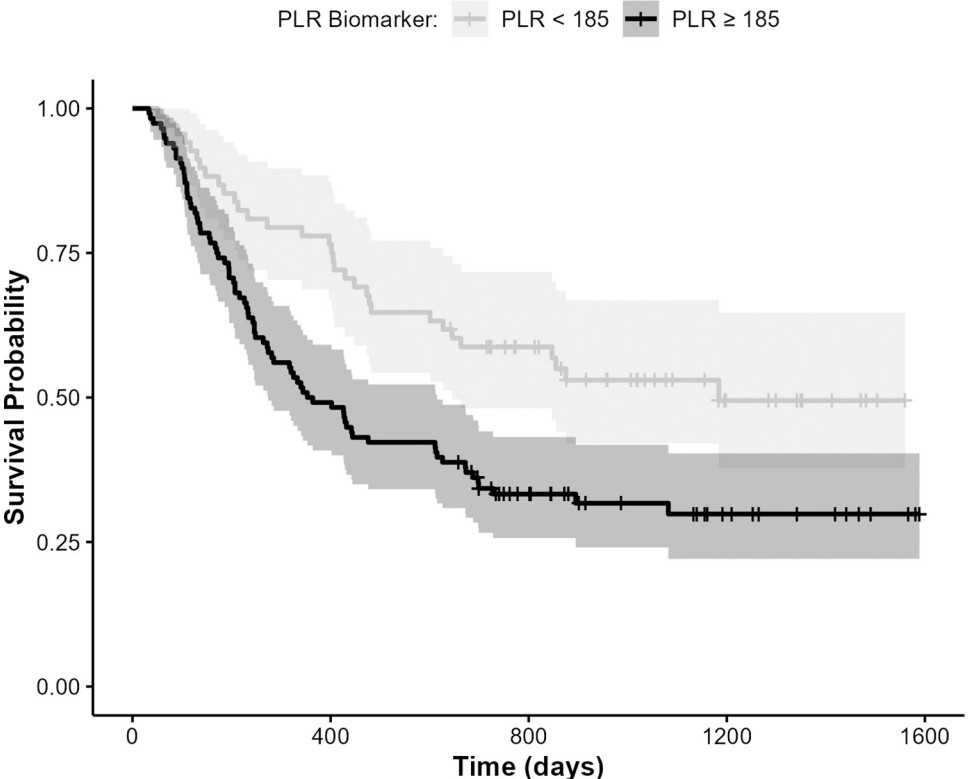

**Fig 3. Kaplan Meier survival curve for NSCLC patients stratified by PLR systemic inflammation, PLR < 185 and PLR ≥ 185 (n = 184), showing worse survival probability (p = 0.005) for patients having greater inflammation (i.e., above PLR cutoff).**

## Conclusions

Systemic inflammatory biomarkers—NLR, PLR, ALI—are known predictors of advanced NSCLC survival [38]. This is supported by new data from a large sample of patients assessed at diagnosis and followed for two years. Novel data show patients' baseline depressive symptoms to be significantly associated with the same biomarker predictors, adjusting for demographic, smoking, and functional status variables. The relationship is strongest for those with moderate to severe depressive symptoms, being 1.3 to 3 times more likely to also have prognostically poor inflammation levels. Biological and psychological domains often viewed as disparate were found to covary in a significant manner, an effect replicated across the three biomarkers.

Immune-based therapies are standard of care in patients with non-small cell lung cancer [23]. A meta-analysis of eight randomized clinical trials demonstrated that the administration of chemotherapy in combination with antibodies that bind to and block the T cell PD-1 and CTLA-4 inhibitory pathways (so called checkpoint inhibitors-CPI) significantly improve over-all and progression-free survival [39]. Inflammatory immune processes could alter and possibly subvert the anti-tumor immune response generated by CPI. Indeed, Ayers et al. (2021) showed elevated NLR to correlate with shorter treatment times and overall survival in lung patients receiving PD-1/PD-L1 inhibitors [33]. A large proportion of patients—35% found here—had significant depressive symptoms prior to receipt of immune-based therapy. The mechanism whereby depression leads to reduced treatment response or poorer survival is likely multi-factorial, but the current literature supports a model in which attendant inflammation exerts an inhibitory effect on immune function. These data support further investigation

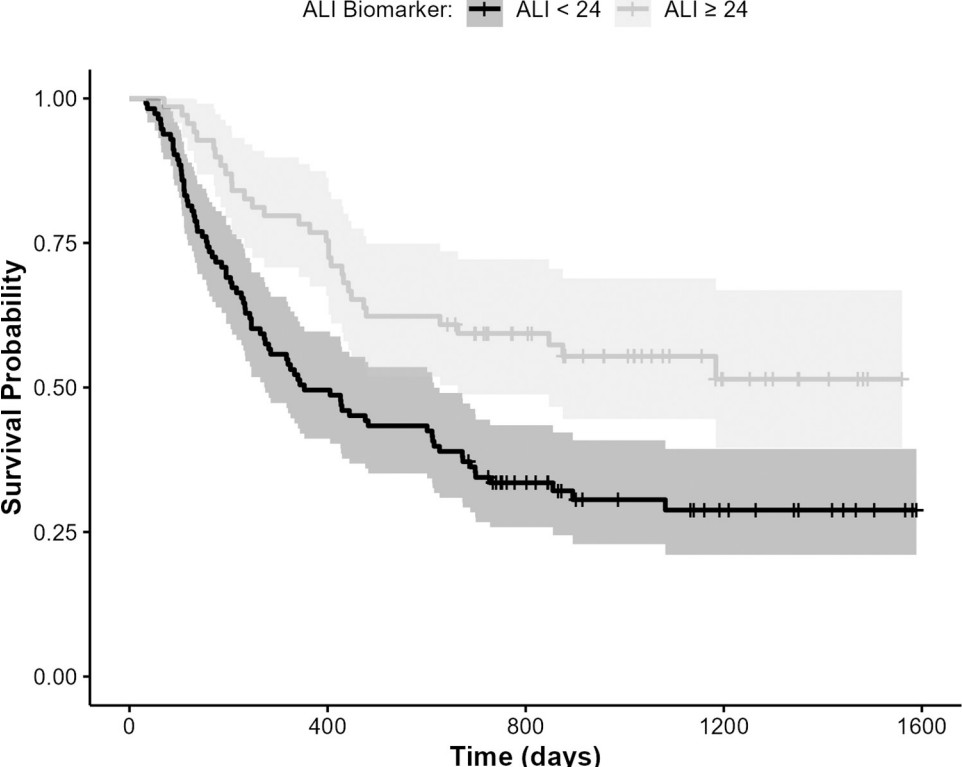

**Fig 4. Kaplan Meier survival curve for NSCLC patients stratified by ALI systemic inflammation, ALI < 24 and ALI ≥ 24 (n = 182), showing worse survival probability (p<0.005) for patients having greater inflammation (i.e., below ALI cutoff).**

of depression and its mechanisms in relationship the efficacy of current therapeutic modalities.

Close inspection of the data highlighted individual differences in the association between depression and inflammation elevations. That is, data show the majority of depressed patients had prognostically poor PLR (Fig 6) and ALI (Fig 7) levels, double to triple the number of persons without depressive symptoms. The numbers are even more compelling when contrasted

**Table 3. Associations between depression symptoms (PHQ-9) and systemic inflammation ratios at baseline.**

| SIR Outcome | PHQ-9 Estimate | 95% CI | *p*-value |
|---|---|---|---|
| NLR | 1.03 | 1.00, 1.05 | 0.02 |
| PLR | 1.02 | 1.00, 1.04 | 0.02 |
| ALI | 0.97 | 0.94, 0.99 | 0.009 |

SIR, systemic inflammation ratio; NLR, neutrophil-lymphocyte ratio; PLR, platelet lymphocyte ratio; ALI, advanced lung cancer inflammation index; CI, confidence interval; PHQ-9, Patient Health Questionnaire-9; ECOG-PS, Eastern Cooperative Oncology Group Performance Scale.

Note: NLR, PLR, and ALI were log-transformed in modeling, so exponentiated estimates are provided. Estimates represent the increase in geometric mean for the corresponding SIR outcome per point increase in PHQ-9. NLR and PLR models were adjusted for ECOG-PS, race, sex, age, marital status, education level (high school or less, beyond high school) and BMI. The ALI model was adjusted for ECOG-PS, race, sex, age, marital status, education level, and treatment; BMI is used to calculate ALI, so the model was not adjusted for BMI.

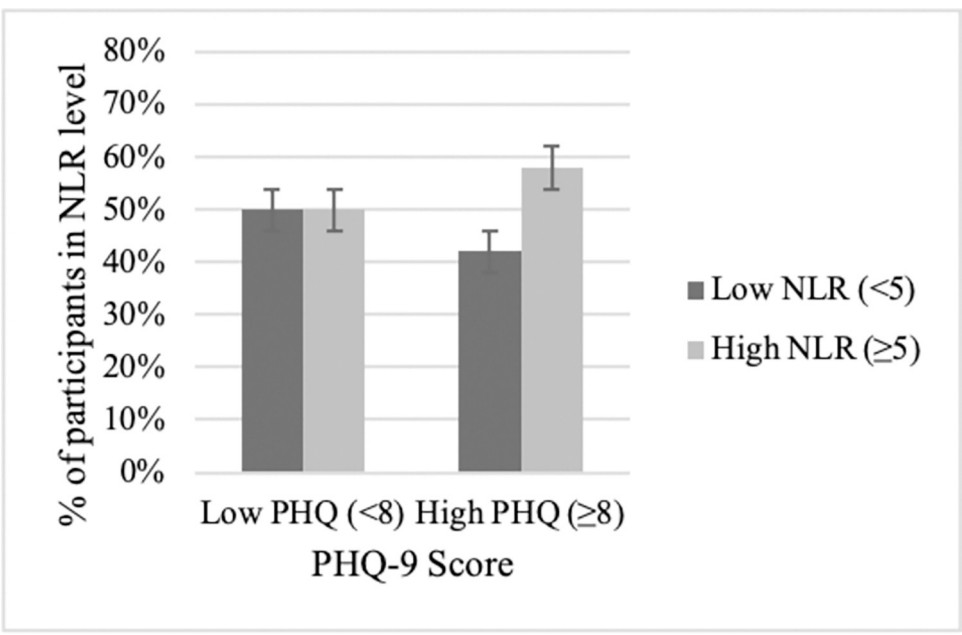

**Fig 5. NSCLC patients classified into PHQ/NLR subgroups.** Patients with no/low depressive symptoms were as likely to have low NLRs (<5; 50%) as have high NLRs (≥5; 50%). In contrast, for patients with moderate/severe depressive symptoms, significantly more patients had prognostically worse, high NLRs (≥5; 58%) rather than low NLRs (<5; 42%). Percentage data are provided with error bars.

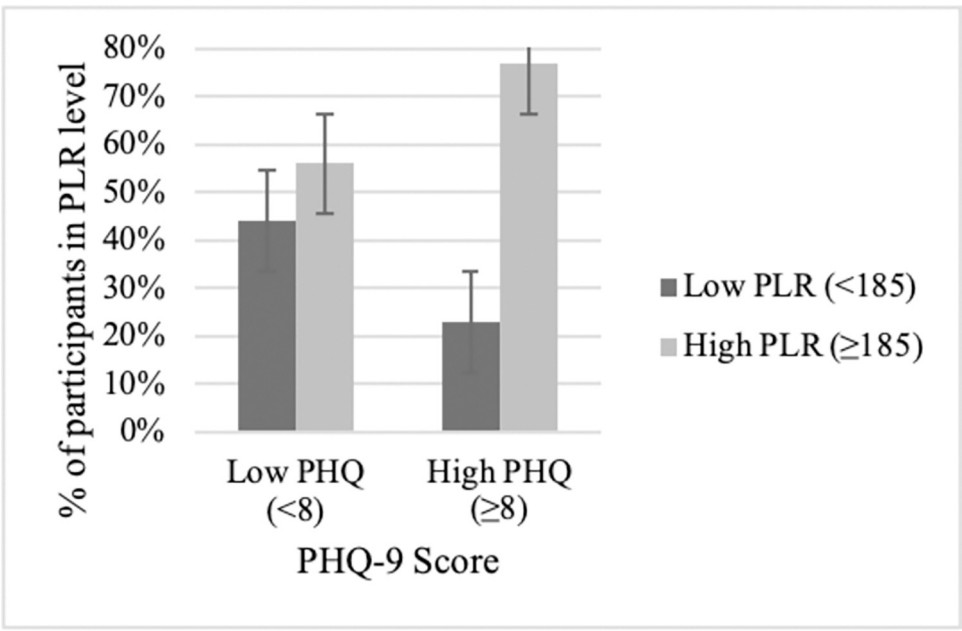

**Fig 6. NSCLC patients classified into PHQ/PLR subgroups.** Patients with no/low depressive symptoms were as likely to have low PLRs as to have high PLRs (left side). In contrast, for patients with moderate/severe depressive symptoms, significantly more patients had prognostically worse, high PLRs rather than low PLRs (right side). Percentage data are provided with error bars.

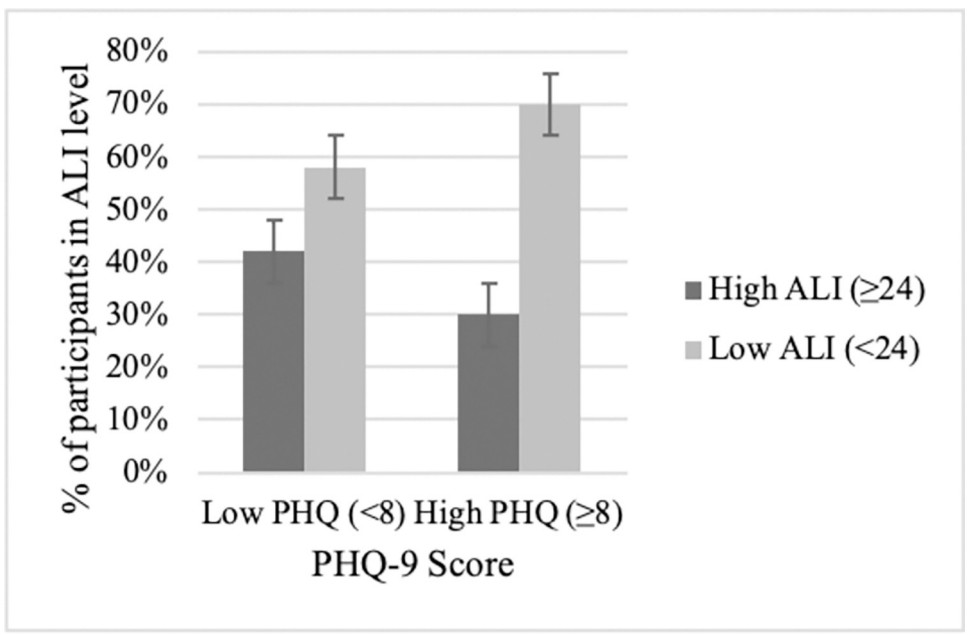

**Fig 7. NSCLC patients classified into PHQ/ALI subgroups.** Patients with no/low depressive symptoms were as likely to have low ALIs as to have high ALIs (left side). In contrast, for patients with moderate/severe depressive symptoms, significantly more patients had prognostically worse, low ALIs rather than high ALIs (right side). Percentage data are provided with error bars.

with currently accepted indicators of poor survival. That is, there were more depressed/elevated inflammation patients in this study than patients with low performance status (ECOG $\geq$ 2; 13%), overweight status (29%), and high school education (34%). Also, depressed NSCLC patients have a constellation of negative emotions and cognitions heightening vulnerability and lessening resilience [40]. For example, they perceive, personally, that little can be done control the disease and are pessimistic about treatment control as well. Theirs is a unique struggle with lung cancer, one different from peers, still concerned but able to cope, but doing so without a depression burden [40].

It is unknown if the depression/inflammation association would remain once cancer treatment begins. However, we have shown the *trajectory* of depression from diagnosis through two years (18 assessments) predicted NSCLC patients' survival (HR = 1.09, 95% CI = 1.03–1.15, p = .002), above and beyond baseline depression, sociodemographics, smoking status, cell type, *and* receipt of targeted treatments and immunotherapies [6]. Taken together, data support psychological, behavioral, and biologic toxicities of depression capable of influencing treatment response and/or survival.

The validity of the depression and SIR associations found here is supported by pathophysiology studies of psychiatric disorders [14,40,41] including mood disorders [15,42,43]. In NLR studies with individuals with depression but not cancer, the relationship between depression severity and NLR elevation is evident. Patients diagnosed with major depressive disorder (MDD) but not yet treated have been found with significant elevations of NLR in comparison to matched controls without MDD, an effect replicated [44–46]. Further, study of major depressive disorder (MDD) "severity" has found patients with MDD who previously attempted suicide had significantly higher NLRs compared to matched MDD patients without a suicide history [47].

The non-cancer studies are important for interpreting the present findings, as in combination, they show a dose response relationship between depression severity ad elevated NLR. This data provides a conceptual replication, i.e. the elevated depression/elevated inflammation effect is the same, but now observed in a cancer sample. PLR and ALI have not been studied in MDD patients, but here the depression/NLR effect was replicated in PLR and ALI analyses, adding reliability and validity the findings. And, it is likely that similar effects would be found with MDD patients. A plausible interpretation of this data is a differential effect of depression severity which "adds" to patients' basal level of inflammation arising from other sources. This may be a contributing mechanism to the uniquely high rates of depression in found LC patients at diagnosis [5,7,8] and the toxicity of the depression trajectory thereafter predicting lower survival [6].

Strengths and weaknesses are considered. Uniform timing of data collection during the difficult diagnostic period was clinically and methodologically important. The sample was at least equal to if not larger than those found in multiple SIR/OS meta analyses [27,29–33,39] and sufficient for this first test of depression in relationship to lung biomarkers [48]. The SIRs are accurate, precise, and robust survival predictors across tumor types [49]. The PHQ-9 has equivalent psychometric strengths [50], and the homogeneity of the sample and the robust range of depressive symptoms found added power. Diagnostic interviews were not done, with the number of patients with MDD unknown. However, MDD criterion symptoms, e.g., low mood, anhedonia, cognitive difficulties, hopelessness, and suicidality, were endorsed by the patients, as has been the case other studies of depressed NSCLC patients [6,9,40,51]. The sample had comparatively low ethnic and racial diversity (18% vs. 22% nationally in the US) [52], potentially limiting generalizability.

In conclusion, new therapies for NSCLC are changing previously bleak, 5-month survival prognosis [53] to 2 years and longer [54]. However, variability in patient response to new therapies exists [2,3], with a pressing need to identify biomarkers predictive of first line therapy failure. The clinical significance of depression/systemic inflammation associations found here require further study as individually, the variables are prognostic for NSCLC survival [6,55,56]. Intensive study of depression among patients with NSCLC, combined with measures of cell biology, inflammation, and immunity, is needed to extend these findings and discover their mechanisms, with the long term aim to improve patients' quality of life, treatment responses, and longevity.

## Supporting information

**S1 Table. Overall p-values from univariate linear analysis.**
(PDF)

**S2 Table. Overall p-values from univariate cox regression.**
(PDF)

## Acknowledgments

We thank the Recruitment, Intervention, and Survey Shared Resource at The Ohio State University Comprehensive Cancer Center for data management, Strategic Research Group for PRO assessments, and John Covarrubias for early assistance.

## Author Contributions

**Conceptualization:** Barbara L. Andersen, William E. Carson.

**Data curation:** John Myers, Tessa Blevins, Rachel M. Smith, Sarah Reisinger.

**Formal analysis:** John Myers.

**Funding acquisition:** David P. Carbone, Carolyn J. Presley, Peter G. Shields.

**Investigation:** Barbara L. Andersen, Kylie R. Park, Sarah Reisinger, William E. Carson.

**Methodology:** Barbara L. Andersen, John Myers, William E. Carson.

**Project administration:** Barbara L. Andersen, Sarah Reisinger.

**Resources:** Kylie R. Park, Rachel M. Smith.

**Supervision:** Barbara L. Andersen, Sarah Reisinger, Peter G. Shields.

**Validation:** Kylie R. Park, Rachel M. Smith.

**Visualization:** John Myers, Tessa Blevins, Kylie R. Park, Rachel M. Smith.

**Writing – original draft:** Barbara L. Andersen, John Myers, William E. Carson.

**Writing – review & editing:** Barbara L. Andersen, John Myers, Tessa Blevins, Kylie R. Park, Rachel M. Smith, Sarah Reisinger, David P. Carbone, Carolyn J. Presley, Peter G. Shields, William E. Carson.

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
