## [Decision Letter · Decision Letter 0]

24 Oct 2022

PONE-D-22-26266Depression in association with neutrophil-to-lymphocyte, platelet-to-lymphocyte, and advanced lung cancer inflammation index biomarkers predicting lung cancer survivalPLOS ONE

Dear Authors,

Thank you for submitting your manuscript to PLOS ONE. After careful consideration, we feel that it has merit but does not fully meet PLOS ONE’s publication criteria as it currently stands. Therefore, we invite you to submit a revised version of the manuscript that addresses the points raised during the review process.

We look forward to receiving your revised manuscript.

Kind regards,

Tomasz Urbanowicz

Academic Editor

PLOS ONE

Journal Requirements:

"PS:The Ohio State University Comprehensive Cancer Center; Pelotonia

DC: The Ohio State University Comprehensive Cancer Center; Pelotonia

"B.L. Andersen, J.Meyers, T. Blevins, K.R. Park, R.M. Smith, S. Reisinger, and W.E. Carson have no competing interests to declare. D.P. Carbone reports personal fees from Abbvie, Adaptimmune, Agenus, Amgen, Ariad, AstraZeneca, Biocept, Boehringer Ingelheim, Celgene, Clovis, Daiichi Sankyo, Inc. (DSI), EMD Serono, Flame Biosciences, Foundation Medicine, G1Therapeutics/Intellisphere, GenePlus, Genentech/Roche, Glaxo-Smith-Kline, Gloria Biosciences, Gritstone, Guardant Health, Humana, Incyte, Inivata, Inovio, Janssen, Kyowa Kirin, Loxo Oncology, Merck, MSD, Nexus Oncology, Novartis, Oncocyte, Palobiofarma, Pfizer, prIME Oncology, Stemcentrx, Takeda Oncology, and Teva; and grants and personal fees from Bristol Myers-Squibb (BMS) outside the submitted work during the conduct of the study. C.J. Presley reports grants from the National Institute of Aging and the National Cancer Institute during the conduct of the study. P.G. Shields reports grants from National Cancer Institute during the conduct of the study."

5. Please upload a copy of Figure 5, to which you refer in your text on page 16. If the figure is no longer to be included as part of the submission please remove all reference to it within the text.

Additional Editor Comments:

Dear Authors,

Please find reviewers’ suggestions and provide point by point answers.

Kind regards

Tom Urbanowicz

Reviewers' comments:

Reviewer's Responses to Questions

**Comments to the Author**

1. Is the manuscript technically sound, and do the data support the conclusions?

Reviewer #1: Partly

Reviewer #2: Partly

2. Has the statistical analysis been performed appropriately and rigorously? 

Reviewer #1: No

Reviewer #2: Yes

3. Have the authors made all data underlying the findings in their manuscript fully available?

Reviewer #1: Yes

Reviewer #2: Yes

4. Is the manuscript presented in an intelligible fashion and written in standard English?

Reviewer #1: Yes

Reviewer #2: No

5. Review Comments to the Author

Reviewer #1: The study aims to assess depression in association with neutrophil-to-lymphocyte, platelet-to-lymphocyte, and advanced lung cancer inflammation index biomarkers predicting lung cancer survival.

The manuscript could be improved based on the following comments.

Measures

Page 8, typo error 'following ace[ted'

Page 8, the sentence ‘income level, and ever smoker (Y/N) as self-reported by patients; lung’ requires revision.

Statistical Analysis

The level of accepted statistical significance to be stated.

The proper citation for the R software to be stated.

Results

The results of identification of potential covariates using correlation or chi-square test to be presented.

Page 12, if 1 decimal point is used for the percentage figure, ensure all are standardized throughout the manuscript. Likewise with 95%CI, two vs three decimal points.

Page 12, for ‘and treatment received.’ it is to be written as ‘and treatment received at baseline.’ (as stated in Table 2 footnote)

Page 13, the word ‘95%CI’ was missing for some CI figures.

Page 13 , for ‘The log rank test yielded p<0.0051.’ is the result correct? Please double-check others as well with Figure 2.

Page 13, log rank for ALI to be stated.

Page 14, the statement on adjustment of covariates in the text and Table 3 footnote is different.

Page 14 Table 3 footnote, typo error ‘covariates save BMI’.

Figure 1, 2 and 3 in the PDF form are not very clear and difficult to visualize (photos in tiff format are fine).

Figure 2, the p value to be denoted in the figure footnote. If the p values here are log rank p values, ensure the p values are similar to those cited in Page 13.

Figure 3 the error bar to be clearly denoted in the figure footnote.

Figure 3 footnote, typo Fig 35B, Fig 35C.

The citations in text and list of references were not conformed to the journal format.

Reviewer #2: Thank you for possibility to review this interesting paper.

The study points out important issues related to problems occurring together with lung cancer diagnosis and treatment.

The paper is well written, though it includes several typos, which should be corrected.

The study design is pretty good, though some details might have been added to strengthen it.

My comments come as follows:

1. the introduction section is too long and contains some parts which further occur in the Discussion. It should be shortened.

2. Methods it is unclear if the patients had reactive depression related to the diagnosis or previously existing, or both subgroups are included. Moreover, it seems that the study comprises patients at different days after getting diagnosis - I suppose that the depression intensity may be different at first days and some people may or not develop improvement or worsening after some period of time

3. My largest concern, and I do not know if it is able to solve that item, is the question if the inflammatory intensity was related to the cancer stage/advancement/influence at other organs, or depression itself. Please comment on this, at least for the reviewer answer

4. do the authors suggest that inflammatory response was the result of depression or depression resulted (In part) from the inflammation.

5. The figures and tables are of poor quality

6. PLOS authors have the option to publish the peer review history of their article (what does this mean?). If published, this will include your full peer review and any attached files.

Reviewer #1: No

Reviewer #2: **Yes: **Anna Olasinska-Wisniewska

---

## [Author Response · Author response to Decision Letter 0]

3 Jan 2023

Response to comments re: Formatting

NOTE: All references to line and page numbers below come from the Revised Article with Changes Highlighted text.

1. We have made every effort to use correct formatting. Author comments/replies are provided in italics.

The manuscript text has been modified as follows (modifications underlined):

“Research personnel completed informed consent face-to-face (verbal with written materials) during a patient’s first clinic appointment with a thoracic oncologist and written informed consent was obtained and the signature witnessed.” See lines 184-186, page 7. 

"PS: The Ohio State University Comprehensive Cancer Center; Pelotonia

DC: The Ohio State University Comprehensive Cancer Center; Pelotonia

Cover letter statements are as follows:

"PS: Pelotonia Grant received from The Ohio State University Comprehensive Cancer Center;

DC: Pelotonia Grant received from The Ohio State University Comprehensive Cancer Center. 

The funders had no role in study design, data collection and analysis, decision to publish, or preparation of the manuscript. No authors received salary for this work.”

"B.L. Andersen, J.Meyers, T. R. Blevins, K.R. Park, R.M. Smith, S. Reisinger, and W.E. Carson have no competing interests to declare. D.P. Carbone reports personal fees from Abbvie, Adaptimmune, Agenus, Amgen, Ariad, AstraZeneca, Biocept, Boehringer Ingelheim, Celgene, Clovis, Daiichi Sankyo, Inc. (DSI), EMD Serono, Flame Biosciences, Foundation Medicine, G1Therapeutics/Intellisphere, GenePlus, Genentech/Roche, Glaxo-Smith-Kline, Gloria Biosciences, Gritstone, Guardant Health, Humana, Incyte, Inivata, Inovio, Janssen, Kyowa Kirin, Loxo Oncology, Merck, MSD, Nexus Oncology, Novartis, Oncocyte, Palobiofarma, Pfizer, prIME Oncology, Stemcentrx, Takeda Oncology, and Teva; and grants and personal fees from Bristol Myers-Squibb (BMS) outside the submitted work during the conduct of the study. C.J. Presley reports grants from the National Institute of Aging and the National Cancer Institute during the conduct of the study. P.G. Shields reports grants from National Cancer Institute during the conduct of the study."

In addition to the text above, the following information has been added and provided in the revision cover letter:

“This does not alter our adherence to PLOS ONE policies on sharing data and materials.”

5. Please upload a copy of Figure 5, to which you refer in your text on page 16. If the figure is no longer to be included as part of the submission, please remove all reference to it within the text.

Previously, 3 figures were provided on a single page. We now provide figures as separate documents (Fig 2-7.tiff). There remains reference to Fig 5 as is it part of the reformatted figure display and numbering. 

Reviewer's Responses to Questions

1. Is the manuscript technically sound, and do the data support the conclusions?

Reviewer #1: Partly

Reviewer #2: Partly

2. Has the statistical analysis been performed appropriately and rigorously? 

Reviewer #1: No

Reviewer #2: Yes

3. Have the authors made all data underlying the findings in their manuscript fully available?

Reviewer #1: Yes

Reviewer #2: Yes

4. Is the manuscript presented in an intelligible fashion and written in standard English?

Reviewer #1: Yes

Reviewer #2: No

It was unclear if we were to respond the above items independently. Responses to the items are reflected in the specific reviewer queries below. We do hope the text below address the tenor of the ratings.

NOTE: All references to line and page numbers below come from the Revised Article with Changes Highlighted text.

Comments of Reviewer #1

The study aims to assess depression in association with neutrophil-to-lymphocyte, platelet-to-lymphocyte, and advanced lung cancer inflammation index biomarkers predicting lung cancer survival. The manuscript could be improved based on the following comments.

1.1 Measures: Page 8, typo error 'following ace[ted'

Typo has been corrected; see line 217, page 8. 

1.2 Page 8, the sentence ‘income level, and ever smoker (Y/N) as self-reported by patients; lung’ requires revision.

New wording is provided: “…income level, and smoking history.” See line 219, page 8.

1.3 Statistical Analysis: The level of accepted statistical significance to be stated. The proper citation for the R software to be stated.

In the Power section, the text reads as follow: “Based on a significance level of alpha = .05…”. See lines 229-230, page 8. 

In the Statistical Analysis section, the text reads as follows: “A p-value of 0.05 was used as a cutoff for significance.” See line 263, page 9.

The R reference is provided [R Core Team (2021). R: A language and environment for statistical computing. R Foundation for Statistical Computing, Vienna, Austria. URL https://www.R-project.org/]. 

1.4 Results: The results of identification of potential covariates using correlation or chi-square test to be presented.

In Tables 2 and 3 the significant covariates are provided, e.g., Table 2 text reads as follows: “Note: Associations are between inflammation biomarkers and overall survival.... The models are adjusted for ECOG-PS (dichotomized as 0-1, 2 or more), age (dichotomized as <60, 60+), educational level (dichotomized as no college education, at least some college), smoking status, and treatment received at baseline.” See lines 374-378, page 12 and lines 656-661, page 14.

Further, two supporting tables (S1-2 Table) provide the analytic results for covariate selection. 

1.5. Page 12, if 1 decimal point is used for the percentage figure, ensure all are standardized throughout the manuscript. Likewise with 95%CI, two vs three decimal points.

We have made every effort to standardize decimal points and CIs throughout the manuscript.

1.6. Page 12, for ‘and treatment received.’ it is to be written as ‘and treatment received at baseline.’ (as stated in Table 2 footnote)

The text has been modified as suggested. See line 317, page 11.

1.7. Page 13, the word ‘95%CI’ was missing for some CI figures. 

All the figures have been revised to include ‘95% CI.’

1.8. Page 13, for ‘The log rank test yielded p<0.0051.’ is the result correct? Please double-check others as well with Figure 2.

The PLR result is correct and but now appears as p=0.005. All values have been checked. See lines 393-394, page 12.

1.9. Page 13, log rank for ALI to be stated.

The log rank data is provided and reads as follows: “The log-rank test indicated a significant difference in survival (p < 0.005) between groups (Fig 4).” See lines 534-535, page 13.

1.10. Page 14, the statement on adjustment of covariates in the text and Table 3 footnote is different.

The wording has been corrected and the specification of covariates appears in the legend of Table 3. See lines 657-661, page 14.

1.11. Page 14 Table 3 footnote, typo error ‘covariates save BMI’.

The wording has been changed and that expression eliminated.

1.12. Figure 1, 2 and 3 in the PDF form are not very clear and difficult to visualize (photos in tiff format are fine). If the p values here are log rank p values, ensure the p values are similar to those cited in Page 13.

P values have been eliminated from the figures (now in TIFF format) and now appear in the Figure titles. 

1.14. Figure 3 the error bar to be clearly denoted in the figure footnote.

Notation now appears in the legends of Figs 6-8

1.15. Figure 3 footnote, typo Fig 35B, Fig 35C.

Typos do not appear. Please note that the prior version placed 3 figures on one page. In this revision the figures are separated, Fig 5-7.

1.16 The citations in text and list of references were not conformed to the journal format.

All citations now follow journal format.

Reviewer #2

Thank you for possibility to review this interesting paper. The study points out important issues related to problems occurring together with lung cancer diagnosis and treatment. The paper is well written, though it includes several typos, which should be corrected. The study design is pretty good, though some details might have been added to strengthen it.

My comments come as follows:

2.1 the introduction section is too long and contains some parts which further occur in the Discussion. It should be shortened.

The prior intro was 3 pages the current is a shortened 2 pages and 1/3. Thank you for this suggestion.

2.2 Methods it is unclear if the patients had reactive depression related to the diagnosis or previously existing, or both subgroups are included. Moreover, it seems that the study comprises patients at different days after getting diagnosis - I suppose that the depression intensity may be different at first days and some people may or not develop improvement or worsening after some period of time.

The comment raises a question about prior depression history; however we were interested in depressive symptoms at diagnosis, per se. The differences in the sample regarding time to assessment are minor. Trajectory data from many cancer groups do show symptom change (declines) once treatment has begun, though not before. Here, all patients were assessed pretreatment.

2.3. My largest concern, and I do not know if it is able to solve that item, is the question if the inflammatory intensity was related to the cancer stage/advancement/influence at other organs, or depression itself. Please comment on this, at least for the reviewer answer.

This is an excellent question. A methodologic strength of the study was the shared (rather than diverse) characteristics of the sample, i.e., all newly diagnosed, same disease stage, same cancer type, uniform assessment pretreatment, and others. This homogeneity increased power. Also, neither depression nor the individual SIRs covaried with the Charleson Comorbidity Index, which quantifies mortality risk due to total comorbidity burden (not just cancer). Also analyses controlled for relevant sociodemographic, disease, and smoking variables which may covary with SIRs.

2.4. do the authors suggest that inflammatory response was the result of depression or depression resulted (In part) from the inflammation.

We offer a plausible interpretation of the data (i.e., depression “adds” to the patients’ basal level of inflammation). We make a comparison of the results with those coming from prior studies with clinically depressed individuals without cancer. We write as follows: 

“The non-cancer studies are important for interpreting the present findings, as in combination, they show a dose response relationship between depression severity and higher NLR. This data provides a conceptual replication, i.e. the elevated depression/elevated inflammation effect is the same, but now observed in a cancer sample. PLR and ALI have not been studied in MDD patients, but here the depression/NLR effect was replicated in PLR and ALI analyses, adding reliability and validity the findings. And, it is likely that similar effects would be found with MDD patients. A plausible interpretation of the data is a differential effect of depression severity which “adds” to patients’ basal level of inflammation arising from other sources. This may be a contributing mechanism to the uniquely high rates of depression in found LC patients at diagnosis [5,7,8] and the toxicity of the depression trajectory thereafter predicting lower survival 6].” See lines 958-966, page 17, and lines 1006-1007, page 18. 

2.5. The figures and tables are of poor quality.

We apologize. All Tables and Figures have been improved.

PLOS authors have the option to publish the peer review history of their article. 

No, we decline this opportunity.

---

## [Decision Letter · Decision Letter 1]

10 Feb 2023

Depression in association with neutrophil-to-lymphocyte, platelet-to-lymphocyte, and advanced lung cancer inflammation index biomarkers predicting lung cancer survival

PONE-D-22-26266R1

Dear AUTHORS,

We’re pleased to inform you that your manuscript has been judged scientifically suitable for publication and will be formally accepted for publication once it meets all outstanding technical requirements.

Kind regards,

Tomasz Urbanowicz

Academic Editor

PLOS ONE

---

## [Editor Report · Acceptance letter]

17 Feb 2023

PONE-D-22-26266R1 

Depression in association with neutrophil-to-lymphocyte, platelet-to-lymphocyte, and advanced lung cancer inflammation index biomarkers predicting lung cancer survival 

Dear Dr. Carson:

I'm pleased to inform you that your manuscript has been deemed suitable for publication in PLOS ONE. Congratulations! Your manuscript is now with our production department. 

Kind regards, 

on behalf of

MR Tomasz Urbanowicz 

Academic Editor

PLOS ONE